# Extraction of coronary thrombus-derived exosomes from patients with acute myocardial infarction and its effect on the function of adventitial cells

Youfu He[1,2,3], Bo Wang[4], Yu Qian[5], Debin Liu[6], Qiang Wu[2,3]*

1 Medical College, Guizhou University, Guiyang, Guizhou Province, China, 2 Department of Cardiology, Guizhou Provincial People's Hospital, Guiyang, Guizhou Province, China, 3 Guizhou Provincial Cardiovascular Disease Clinical Medicine Research Center, Guiyang, Guizhou Province, China, 4 Department of Urology, Guizhou Provincial People's Hospital, Guiyang, Guizhou Province, China, 5 Department of Cardiology, The Second Affiliated Hospital of Zunyi Medical University, Guiyang, Guizhou Province, China, 6 Department of Cardiology, The Second People's Hospital of Shantou, Shantou, Guangdong Province, China

☯ These authors contributed equally to this work.

* wuqiang@gz5055.com

**Data Availability Statement:** The TE-related gene expression dataset was obtained from the Gene Expression Omnibus (GEO) database (https://www.

## Abstract

### Background

Type I acute myocardial infarction (T1MI) has a very high morbidity and mortality rate. The role of thrombus-derived exosomes (TEs) in T1MI is unclear.

### Methods

The objective of this study was to identify the optimal thrombolytic drug and concentration for extracting TEs. To this end, a series of time and concentration combinations were tested. Subsequently, the effect of TEs on thrombus-adjacent cells was investigated. Finally, we conducted lncRNA microarray analysis on the extracted TEs (GSE213115).

### Results

TEs has been demonstrated to promote necrosis, autophagy, and ferroptosis of human cardiomyocytes, while inhibiting the proliferation and migration of human umbilical vein endothelial cells (HUVECs). Furthermore, TEs can stimulate the proliferation and migration of smooth muscle cells, and induce a transformation from a contractile to a secretory phenotype. Bioinformatics analysis revealed that five lncRNAs, AC068418.2, AC010186.3, AL031430.1, AC121333.1, and AL136526.1, exhibited significant differential expression in TE and regulated cell autophagy and ferroptosis by directly binding to TP53, TP63, and RELA, respectively.

ncbi.nlm.nih.gov/geo/) under accession number (GSE213115).

**Funding:** This work was supported by the [Guizhou Provincial Health and Wellness Commission] under Grant [gzwkj 2021-102]; The National Natural Science Foundation of China (No. 82260084); Guizhou Provincial Science and Technology Agency Project (Qian Ke He Foundation ZK [2023] General 217); and Support by Key Advantageous Discipline Construction Project of Guizhou Provincial Health Commission in 2023. The funders had no role in study design, data collection and analysis, decision to publish, or preparation of the manuscript.

**Competing interests:** The authors have declared that no competing interests exist.

## Conclusions

We demonstrate that TEs as a potential target and research direction for the treatment of heart failure after T1MI. TEs may regulate ferroptosis and autophagy in thrombus-adjacent cells through the enrichment of certain lncRNAs.

## Introduction

Type I myocardial infarction (T1MI) due to coronary thrombosis has a very high morbidity and mortality rate, which has a significant impact on the healthy lifespan and quality of life of human beings [1,2]. MI is a disease caused by local myocardial ischaemia due to rupture of atherosclerotic plaques in coronary arteries or spasm of coronary arteries, and is one of the major causes of death due to the disease worldwide today [3]. Given that thrombosis is a highly rapid process, early diagnosis and treatment of T1MI is of particular importance. However, research and understanding of etiology underlying T1MI, namely coronary thrombosis, is currently limited.

The exosomes are 30–200 nm in size. It belongs to a kind of extracellular vesicle secreted by cells, which is a capsule-like substance with a phospholipid bilayer structure secreted by living cells and is rich in various proteins and genetic materials [4]. Numerous studies have shown that exosomes secreted by various cells are involved in thrombosis. Several studies have confirmed that atherosclerotic (AS) plaques contain high levels of vesicles [5]. These vesicles can affect endothelial function and the accumulation of lipids and leukocytes in the intima by participating in plaque formation and can even affect the stability of coronary plaques and induce thrombosis [6,7]. Previously, the thrombus was considered a waste product in the human vasculature and was the main cause of T1MI. However, little consideration has been given to the effects of these components on the adjacent cells or distal cardiomyocytes.

Several studies have shown that circulating exosomes carry substances that regulate the apoptosis and necrosis of cardiomyocytes following AMI [8,9]. Moreover, insufficient evidence indicates that vascular endothelial and vascular smooth muscle cells are also affected by circulating exosomes and undergo considerable functional changes after AMI onset [9–11]. Several recent studies have shown that many lncRNAs are involved in the development of cardiac diseases by using exosomes as carriers [6]. For example, lncRNAs such as fetal lethal non-coding developmental regulatory RNA (FENDRR) [12], ZFAS1 [13], and NEAT1 [14] are widely involved in various pathophysiological processes such as cardiomyocyte formation, AMI, and heart failure. A wealth of evidence from studies on a diverse range of diseases indicates that an understanding of the function of exosomes and their contents is of paramount importance for the prevention and treatment of disease [15]. At present, no studies have reported on whether exosomes exist in the coronary thrombi or are involved in the regulation of cardiomyocytes after acute myocardial infarction(AMI).

Previous studies of T1MI have focused mainly on aspects such as myocyte necrosis and endothelial cell dysfunction in the region of myocardial infarction. In fact, as early as 2007, Leroyer et al. [5] identified vesicle-like material in cells or tissues adjacent to coronary arteries. In this study, we improved on the technique published by Vella et al. to lyse brain tissue using collagenase and extract exosomes [16]. After improving the lysis protocol, we successfully extracted and characterized exosomes from a coronary thrombus using ultracentrifugation. To our knowledge, this is the first describe, extract, and characterize TEs. Our TE extraction technology was granted a national invention patent in China (patent No. ZL202110554258.6).

This study aimed to provide a new direction for the prevention and treatment of T1MI caused by intracoronary thrombi by further exploring the function of exosomes in these thrombi.

## 2. Materials and methods

### 2.1. Reagents and chemicals

Phosphate-buffered saline (PBS) and DiI were purchased from Thermo Fisher Scientific. Trypsin (BS130) and type II collagenase (BS164) were purchased from BIOSHAP (China). Calcium chloride (499609) was purchased from Sigma–Aldrich. The antibodies αSMA (R25049), CD63 (R23851), GAPDH (380626), GPX4 (381958), LC3A/B (306019), Proliferating Cell Nuclear Antigen (PCNA) (R25294), osteopontin (OPN; 340690) were obtained from ZEN BIOSCIENCE, China. Heparin was purchased from Hainan General Alliance Pharmaceutical Co., Ltd. (China). Urokinase was purchased from Shanghai Tianshili Pharmaceutical Co. Ltd., China. Alteplase was purchased from Boehringer Ingelheim Pharma GmbH and Co. (Denmark). Streptokinase was purchased from Beijing Sihuan Biopharmaceutical Co., Ltd. (Beijing, China). CD34 (ab81289), cTnI (ab209809), caspase8 (ab108333), caspase9 (ab32539), cleaved-caspase3 (P17) (ab32042), SQSTM1/P62 (ab207305), cyclinD1 (ab16663) were purchased from Abcam, USA. The antibodies goat anti-rabbit IgG H&L (HRP) (ZB-2301), goat anti-mouse IgG H&L (HRP) (ZB-2305), and hematoxylin stain (ZLI-9610) were purchased from Solelybio, China. The DAB chromogenic kit (CW0125) and neutral resin (CW0136) were purchased from CWESTERN BLOTIO, China. Polyvinylidene difluoride membranes (IPVH00010) were purchased from Merck Millipore Ltd. (Darmstadt, Germany). Apoptosis kits, including Annexin V-fluorescein isothiocyanate (FITC) and Proliferous Index (PI) (KGA108), were purchased from KeyGEN Biotech (China). Cell cycle assay kit, including PI and RNase A (C1052) was from Byotime, China. ROS assay kit, including DCFH-DA and positive control (S0033S) was from Byotime, China. Exosome-free FBS (abs993) was purchased from Absin (Shanghai, China).

### 2.2. Thrombus tissue selection and preparation

A coronary thrombus is an intracoronary thrombus in patients with T1MI diagnosed using coronary angiography at our hospital's Department of Cardiovascular Medicine (Guizhou Provincial People's Hospital). The thrombi were removed by catheter aspiration and then placed in centrifuge tubes containing 5 ml of PBS and stored at 4˚C for no more than 3 days.

Venous thrombosis occurs in patients with acute lower extremity thrombosis diagnosed by angiography at our institution and whose onset occurs within 24 hours. Venous thrombi removed by thromboaspiration were also placed in PBS and stored briefly at 4˚C.

### 2.3. Serum selection and preparation

Five milliliters of whole blood from the AMI patients was placed in a procoagulation tube, and the serum was extracted by centrifugation at 3000 rpm for 15 min [17]. The serum was stored at -20˚C for no more than 6 months.

### 2.4. Exosome enrichment from thrombus

Different concentrations of type II collagenase (10–200 U/ml) and different concentrations of trypsin (10–200 μg/ml) were added to the coronary thrombi described above. According to the instructions for type II collagenase, 5 mM calcium chloride was added to the corresponding groups as an activator of the enzyme. Thereafter, the mixture was placed on a thermostatic shaker at 37˚C and left for an appropriate time (15 min to 1 h) until the thrombus was

completely dissolved. The mixture obtained above was centrifuged in a high-speed centrifuge at 4˚C and 300 × g for 5 minutes to remove incompletely dissolved thrombus and adherent intimal tissue. Thereafter, dead cells and cell debris were removed by centrifugation at 4˚C and 2000 × g for 10 min. After centrifugation at 4˚C and 100,000 × g for 70 minutes, the precipitate is the exosome [18]. It was also possible to perform centrifugation at 100,000 × g for 70 min to purify the resulting exosomes.

## 2.5. Exosome enrichment from serum

The patient's serum obtained above was placed in a high-speed centrifuge and centrifuged at 4˚C and 2000 × g for 10 minutes to remove dead cells and cell fragments. After centrifugation at 4˚C and 100,000 × g for 70 minutes, the precipitate is the exosome [17].

## 2.6. Western blot

An equivalent of 20 μg total exosome protein was separated by 6–12% sodium dodecyl sulfate-polyacrylamide gel electrophoresis, and then, transferred onto a polyvinylidene difluoride membrane. After blocking with Protein Free Rapid Block buffer, the protein was incubated with the primary antibodies at 4˚C overnight. The next day, the proteins were washed three times with Tris-buffered saline +0.1% Tween-20 (TBST) and incubated with horseradish peroxidase (HRP)-conjugated anti-rabbit IgG secondary antibodies for 2 h at room temperature [19]. The immune complex was assessed using an enhanced chemiluminescence system (P36599; Millipore, Boston, MA, USA), and the intensity of the bands was analyzed using the ImageJ software (Bethesda).

## 2.7. Exosome electron microscopy

Resuspend an appropriate amount of separated exosomes with PBS (10–20ul), take droplets, and place them on the glow-discharged 300-mesh heavy-duty carbon-coated formvar Cu grids (ProSciTech, Kirwan, QLD, Australia) for 5 min, and excess was blotted on filter paper (Whatman, Maidstone, UK). The grids were then washed twice with Milli-Q water and negatively stained with 2.5% uranyl acetate. All samples were analyzed using an H-7650 electron microscope at 100KV [20].

## 2.8. Thrombus electron microscopy

The obtained coronary thrombus was fixed with 4% paraformaldehyde for 48 h. We used 1% osmium tetroxide for the fixation. After post-fixation, all the specimens were dehydrated using a graded series of ethanol. Ultrathin sections 100 nm thick were cut for electron microscopy analysis, set on single slot or 200-mesh copper or nickel grids, and imaged using an electron microscope [21].

## 2.9. Nanoparticle tracking analysis

We measured exosome particle size and concentration using nanoparticle tracking analysis (NTA) at VivaCell Bioscience with ZetaView PMX 110 (Particle Metrix, Meerbusch, Germany) and the corresponding software ZetaView 8.04.02. Isolated exosome samples were appropriately diluted using 1X PBS buffer (Biological Industries, Israel) to measure particle size and concentration. NTA measurements were recorded and analyzed at 11 positions. The ZetaView system was calibrated using 110 nm polystyrene particles. The temperature was maintained at around 23˚C to 30˚C [22].

## 2.10. Cell Counting Kit-8 assay

Cell proliferation was assessed using a Cell Counting Kit-8 (CCK-8) (Beyotime, China). The cells ($2 \times 10^3$ cells/mL) were seeded into a 96-well plate. An appropriate amount of each group of exosomes was co-cultured with the cells for 24 h. After mixing 10μL of CCK-8 solution into each well, the cells were incubated for 2 h at 37°C [19]. Optical density was measured at 450 nm using a microplate reader (Bio-Rad Laboratories, USA) at room temperature.

## 2.11. Cell culture and co-culture with exosomes

Human umbilical vein endothelial cells (HUVEC, Cat.NO. PCS-100-013™) and vascular smooth muscle cells (SMCs, Cat.NO. CRL-1999) were purchased from ATCC (Manassas, VA, USA). Human cardiac myocytes (HCM; cat. 6200) was purchased from ScienCell (USA). HCM and SMC were cultured in DMEM (Cat. C11995500BT, Gibco, USA) containing 5% fetal bovine serum (Cat.NO.10099-141, Gibco, USA). HUVEC was cultured using 1640 medium (Cat.NO. C11875500BT; Gibco) containing 5% fetal bovine serum. All cells were changed once every three days. The cells were grown to 80% confluence and were ready for passaging or use. All exosomes(All from patients with T1MI as previously described or healthy individuals used as controls) were diluted to the appropriate concentration using PBS and co-cultured with cells grown to 80% confluency [23]. When exosomes were co-cultured with the cells, FBS was replaced with EXO-FREE FBS.

## 2.12. Immunofluorescence assay

Exosomes (100ug) were stained with 1μM DiI and incubated at 37°C in the dark for 30 minutes. The mixture was then centrifuged at 120,000 × g for 70 min. After carefully absorbing the supernatant, 1 ml of complete medium containing 10% FBS was added, blown, mixed well, added to the cell culture dish, and co-cultured with the corresponding cells for 24 h. The cells were then fixed with 4% paraformaldehyde, nonspecific antigens were blocked with goat anti-serum, and the cells in each group were incubated overnight with the corresponding marker antibody. The next day, FITC fluorescent label was added to the secondary antibody and incubated in the dark for 1 h. After DAPI staining, the fluorescence was detected using a fluorescence microscope [24].

## 2.13. Flow cytometric analysis

A total of $2 \times 10^5$ cells were incubated at 4°C away from light for 1 h with antibodies targeting the Annexin V-FITC, PI, DCHF-DA (Byotime, Cat.NO:S0033S). After washing twice with PBS, the cells were subjected to a flow cytometric analysis (Beckman Coulter, Fullerton, CA, United States) [25].

## 2.14. Data acquisition

The TE-related gene expression dataset was obtained from the Gene Expression Omnibus (GEO) database (https://www.ncbi.nlm.nih.gov/geo/) under accession number (GSE213115). The microarray dataset contained serum exosome data from three healthy individuals, four T1MI patients, and thrombus exosome data from four T1MI patients, which were tested using the [HTA-2_0] Affymetrix human transcriptome array 2.0 [transcript (gene) version]. Notably, the average gene expression was determined if multiple probes mapped to the same gene.

## 2.15. Identification of differentially expressed LncRNAs (DELncRNAs)

Limma [26], a Bioconductor package in R software, was applied to identify DELncRNAs with thresholds of P-value < 0.05, and | log2FC |> 1.0. Volcano plots and heat maps of the DELncRNAs were constructed using R software.

## 2.16. Identification of LncRNAs associated with ferroptosis and autophagy

mRNAs that directly bound to the screened lncRNAs were obtained from RNAINTER (http://www.rnainter.org/search/) [27]. Data on 565 ferroptosis-related genes were obtained from FerrDb (http://www.zhounan.org/ferrdb/current/) [28]. Autophagy-associated genes were obtained from HADb (http://autophagy.lu/) [29], and 232 autophagy-associated genes were downloaded.

## 2.17. Selection of common genes (CGs)

CGs were obtained by screening mRNAs that could directly bind to previous DELncRNAs and crossing them with autophagy-related or ferroptosis-related genes. Venn diagrams were constructed to visualize the overlap of autophagy- or ferroptosis-related genes with mRNAs using the R package "VennDiagram" (v 1.7.3) [30].

## 2.18. Ethics approval and consent to participate

The ethical review of This study was approved by the Ethics Committee of the Guizhou Provincial People's Hospital (Lun Audit (Research)2022-79). The biological samples used in this study were obtained from patients at the Guizhou Provincial People's Hospital. Recruitment of patients for this study began on 2022/08/01 and ended on 2022/09/01.Three patients diagnosed with T1MI by emergency coronary angiography from the Department of Cardiology of Guizhou People's Hospital were recruited in the preliminary phase of this study. Written informed consent was obtained from the patients during the preoperative interview to facilitate the extraction of coronary thrombus and peripheral blood. All patients were provided with written formal consent prior to the acquisition of samples.

All procedures were performed in accordance with the ethical standards of the responsible committee on human experimentation (institutional and national) and the Helsinki Declaration of 1975, as revised in 2000 (5). Informed consent was obtained from all patients for inclusion in the study.

## 2.19. Statistical analysis

SPSS19.0 and GraphPad 7 were employed for analysis and image rendering of the experimental data, respectively. Comparisons between groups were performed using independent t-tests, those among multiple groups were performed using one-way analysis of variance (ANOVA), and post-pairwise comparisons were performed using LSD-t. Multitime expression profiles were analyzed by repeated measures ANOVA, and post hoc tests were conducted using the Bonferroni correction. P < 0.05 indicated a statistically significant difference.

# 3. Results

## 3.1. Microscopic reveals a vesicle-like exosomes in coronary thrombosis

To obtain a complete morphology of coronary and venous thrombus, we examined coronary and venous thrombosis using electron microscopy techniques. The difference between the two thrombi was observed in the transmission electron microscopy results. Aggregated platelets

and a large amount of fused fibrin with a small amount of sac-like material were observed in coronary thrombus (**Fig 1A**). Their size (approximately 200 nm) is much smaller than that of normal platelets (2–4 mm). Based on its size and shape (concave disk in the middle)[31], we concluded that it was an exosome. As a control, we observed many fibrin fusions, a small number of platelet bundles, and some cells of unknown composition in venous thrombi, and no clear structure resembling an exosome was observed (**Fig 1B**).

In summary, we found the presence of vesicle-like material suspected to be exosomes in coronary thrombosis.

## 3.2. Lysis of thrombus and extraction of TEs using trypsin combined with type II collagenase

We hypothesized that the previously observed vesicle-like material was exosomes. To extract these exosomes, we screened several drugs containing pancreatic enzymes, such as collagenase, alteplase, heparin, urokinase, and streptokinase, which can dissolve or digest thrombi.

To extract the exosomes from the thrombi, we first lysed them. Combined with the previous method of extracting tissue exosomes by lysing brain tissue with collagenase proposed by Vella et al. [16], we screened a combination of different doses of collagenase and trypsin to lyse the thrombus, extracted exosomes separately, and tested their efficiency. The experiment was divided into five main groups: ① group: low-dose trypsin (10μg/ml) + low-dose type II collagenase (10U/ml) group; ② group: low-dose type II collagenase (10U/ml) + high-dose trypsin group (200μg/ml); ③ group: Medium-dose trypsin (20μg/ml) + Medium-dose type II collagenase group (50U/ml), ④ group: High-dose type II collagenase (200U/ml) + High-dose trypsin (200U/ml) group, ⑤ group: High-dose type II collagenase (200U/ml) + Low-dose trypsin (10μg/ml) group, total 5 groups. Thrombi were removed using forceps at 0, 15, 30, 45, and 1 h, weighed, and photographed. The effects of lysis are shown in Fig 2. The lysis efficiency of groups ②, ③, and ④ was significantly better than that of groups ① and ⑤ (Table 1 and Fig 2A and 2B). Ultracentrifugation was then used to extract exosomes from the lysate mixture and identify their surface marker CD63 using Western blotting. The results were positive (Fig 2C and 2D), but the CD63/GAPDH ratio in groups ② and ③ was significantly higher than that in the other groups ($P < 0.05$), which may be related to damage to the exosomal

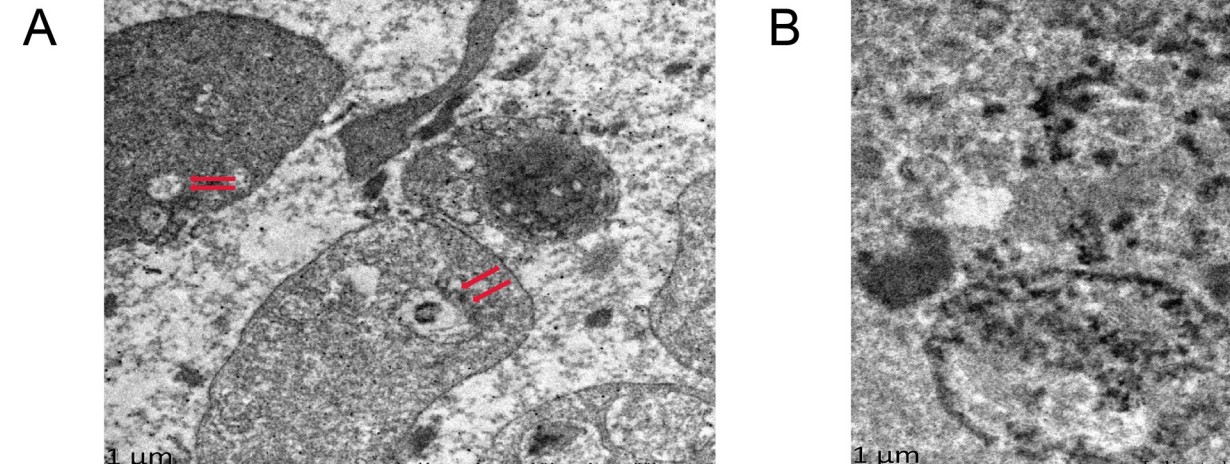

**Fig 1. Microscopic features of thrombus (Transmission electron microscope).** (A) Coronary thrombus: Very small amount of platelets as well as a large amount of fibrin fusion within the thrombus, with a small amount of vesicle-like material in the middle (red arrow), about 200 nm in size. (B) Venous thrombus: More platelet aggregation, disordered fibrin aggregation, no obvious vesicle-like material is seen.

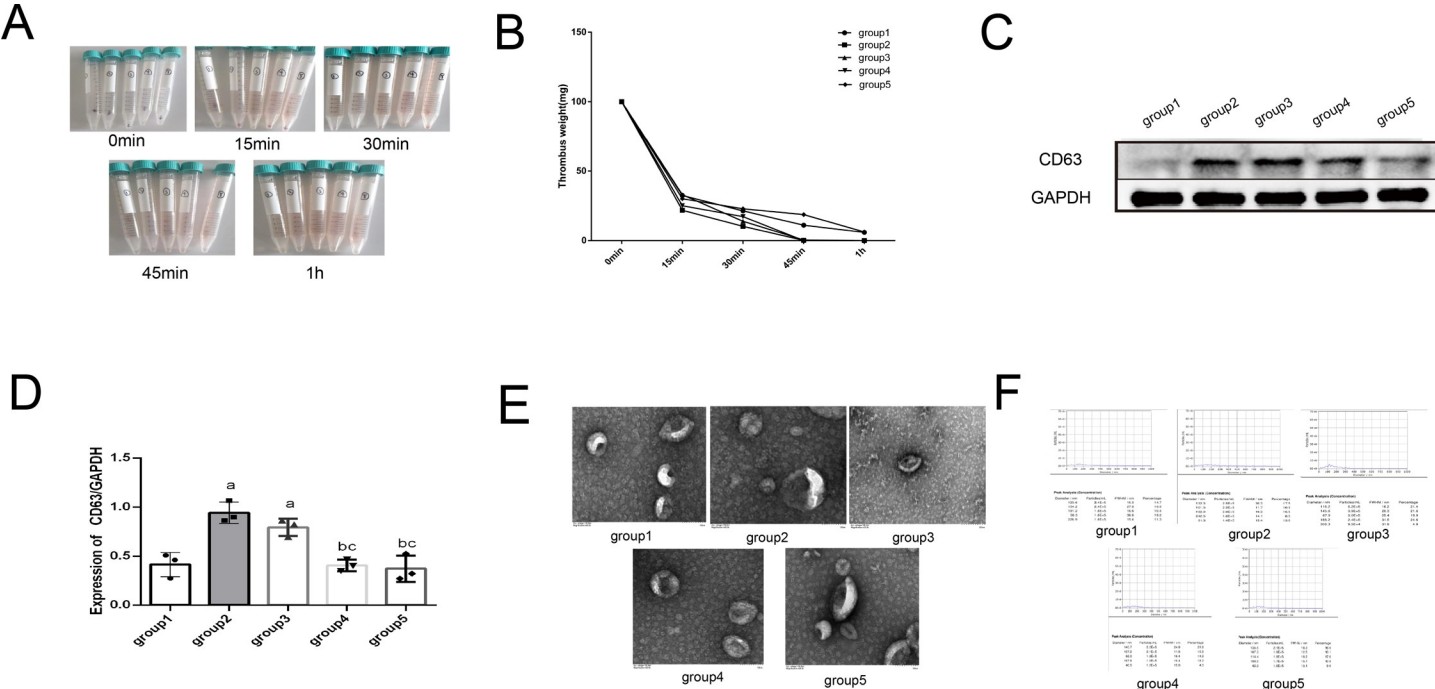

**Fig 2. Different concentrations of collagenase + trypsin dissolved thrombus and extracted exosomes for identification.** (A) Thrombolysis process in each group. (B) Statistics of residual mass in different periods after thrombolysis. (C)The expressions of CD63 and GAPDH in exosomes were detected by Western blot. (D) Western blot result of the expressions of CD63 and GAPDH in exosomes. a: $P<0.05$, compared with group 1. b: $P<0.05$, compared with group 2.c: $P<0.05$, compared with group 3.d: $P<0.05$, compared with group 4. (E)Electron microscopic results of exosomes extracted from dissolved products in each group. (F) NTA results of exosomes in each group.

membrane caused by the excessive dose of trypsin. The morphology of each group of exosomes was examined by electron microscopy. The results showed (Fig 2E) that the exosomes of each group were in the shape of a standard disc with a depression in the middle and a size was 50–200 nm, which was consistent with the morphological characteristics of the exosomes [31]. The size distribution of the exosomes in each group was determined using NTA. The results showed (Fig 2F) that the distribution of exosomes in each group was uniform, and the peak was within 200 nm, which is consistent with the size characteristics of the exosomes [31]. Therefore, combined with the above results, we selected the medium-dose trypsin + type II collagenase group (20 µg/ml trypsin + 50 U/ml collagenase) as the final protocol. It is worth mentioning that we found that a high dose of trypsin treatment may decrease the rate of exosome acquisition (as shown by the decrease of CD63 expression).

**Table 1. Statistical table of a residual mass after thrombolysis in each group (n = 3, $\bar{x} \pm S$).**

|       | group1    | group2           | group3        | group4       | group5              |
|-------|-----------|------------------|---------------|--------------|---------------------|
| 0min  | 100       | 100              | 100           | 100          | 100                 |
| 15min | 32.5±10.2 | 21.8±12.9        | 33±13.3       | 25.1±13.2    | 30.1±11.8           |
| 30min | 21.5±11.1 | 10.3±4.4         | 13.9±9.9      | 17.2±5.9     | 22.9±9.9            |
| 45min | 11.1±4.9  | 0.2±0.01[a]      | 0.5±0.1[a]    | 0[a]         | 18.7±4.8[abcd]      |
| 1h    | 5.9±3.2   | 0[a]             | 0[a]          | 0[a]         | 6.2±3.3[bcd]        |

a:$P<0.05$, compared with group 1. b:$P<0.05$,compared with group 2. c:$P<0.05$,compared with group 3. d:$P<0.05$,compared with group 4.

To further explore the optimal combination of thrombolytic agents, we evaluated the in vitro thrombolytic efficiency of commonly used anticoagulant/thrombolytic drugs (heparin, urokinase, alteplase, and streptokinase) to identify a combination of drugs that would dissolve the thrombus with minimal effect on the purification of exosomes. The experiment was divided into seven groups: ① Trypsin group (50 μg/ml),② type II collagenase group (75 U/ml),③ heparin group (4000 U),④ alteplase group (15 mg),⑤ trypsin (20 μg/ml) + collagenase group (50 U/ml),⑥urokinase group (1.5 million U),⑦ streptokinase group (2 million IU). All clinical doses were in accordance with relevant Chinese guidelines and drug instructions. We also measured the mass of the remaining thrombus at different time points (Fig 3A and 3B, Table 2). As shown in the results, group ⑤ was significantly better than the other six groups ($P < 0.05$). Western blotting was subsequently used to identify the exosome surface marker CD63(Fig 3C and 3D). As shown in the results, CD63 expression was the strongest in the group ②, and the CD63 / GAPDH ratio was much higher than those of the other groups ($P < 0.05$). We also examined the electron microscopy and NTA results of the four groups. Collagenase only, trypsin only, alteplase, and trypsin+alteplase groups. According to the electron microscopy results (Fig 3E), the shape and size of the exosomes were uniform in all groups, which is consistent with the characteristics of exosomes [31]. From the NTA results (Fig 3F), the peaks of all four groups of exosomes were within 200 nm, with a uniform particle size distribution, which was consistent with the characteristics of exosomes [31]. Combined

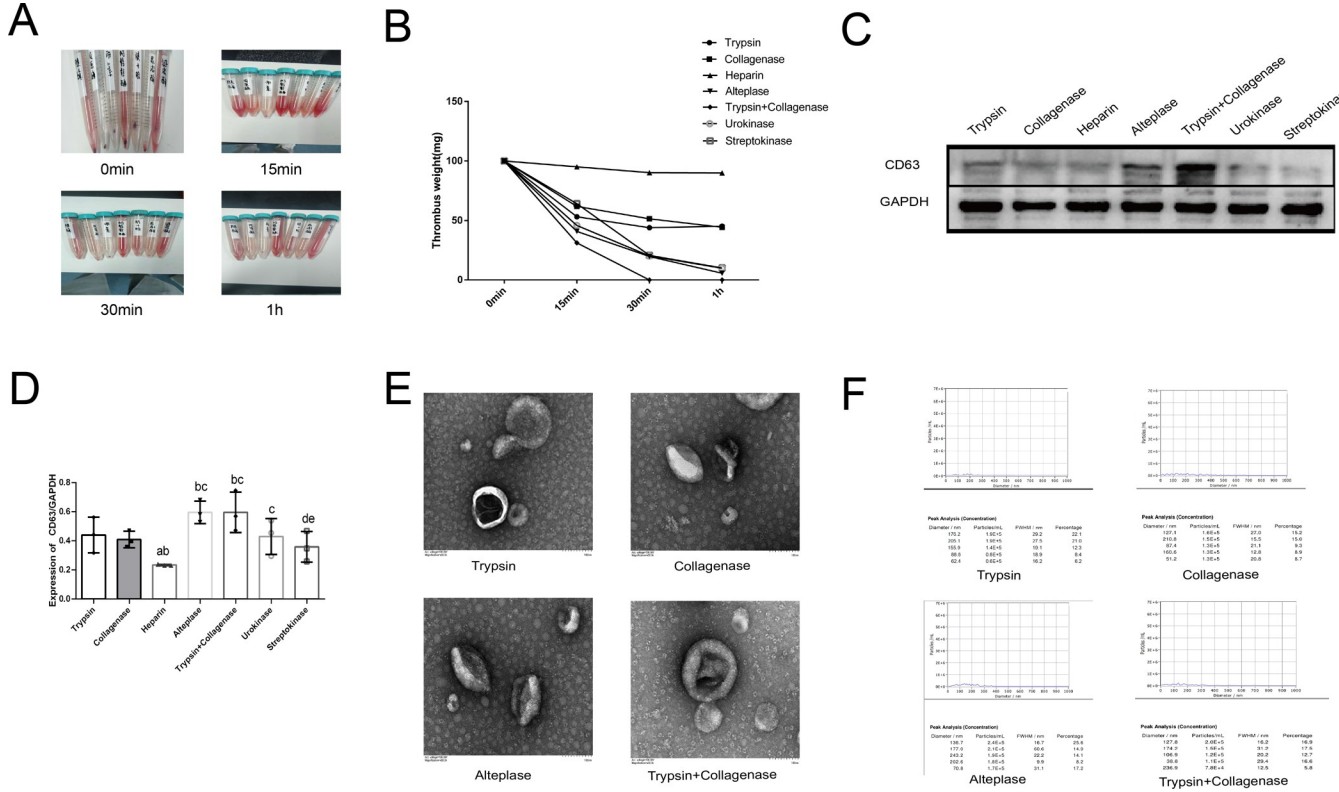

**Fig 3. Different kinds of thrombolytic drugs were used to dissolve thrombus and extract exosomes for identification.** (A)Thrombolysis process in each group. (B) Statistics of residual mass in different periods after thrombolysis. (C)The expressions of CD63 and GAPDH in exosomes were detected by Western blot. (D) Western blot result of the expressions of CD63 and GAPDH in exosomes. a:$P< 0.05$, compared with Trypsin group (50 μg/ml). b: $P<0.05$, compared with type II collagenase group (75 U/ml). c: $P<0.05$, compared with the heparin group (4000 U). d: $P<0.05$, compared with the alteplase group (15 mg). e: $P<0.05$, compared with trypsin (20 μg/ml) + collagenase group (50 U/ml). e: $P<0.05$, compared with the urokinase group (1.5 million U). (E)Electron microscopic results of exosomes extracted from dissolved products in each group. (F) NTA results of exosomes in each group.

**Table 2. Statistical table of a residual mass after thrombolysis in each group (n = 3, $\bar{x} \pm S$).**

| | Trypsin | Collagenase | heparin | Alteplase | Trypsin + Collagenase | urokinase | Streptokinase |
|---|---|---|---|---|---|---|---|
| 0min | 100 | 100 | 100 | 100 | 100 | 100 | 100 |
| 15min | 53.2±4.8 | 61.9±8.1 | 95.1±4.1[ab] | 40.9±3.9[abc] | 31.2±2.2[abc] | 45.9±6.6[bce] | 64.1±8.2[acdef] |
| 30min | 44±6.9 | 51.5±6.5 | 90.2±1.2[ab] | 19.8±4.8[abc] | 0[abcd] | 20.3±4.7[abce] | 20.7±3.9[abce] |
| 1h | 45.2±2.1 | 44.4±5.8 | 90±0.5[ab] | 5.7±1.1[abc] | 0[abcd] | 9.9±3.2[abce] | 10.2±4.4[abce] |

a:P<0.05, compared with Trypsin group (50μg/ml). b:P<0.05, compared with type II collagenase group (75 U/ml). c:P<0.05, compared with heparin group(4000U). d: P<0.05, compared with alteplase group(15mg). e:P< 0.05, compared with trypsin(20μg/ml) + collagenase group (50 U/ml). f: P<0.05, compared with the urokinase group (1.5 million U).

with the expression intensity of CD63, the alteplase and trypsin + collagenase groups were significantly better than the collagenase group ($P < 0.05$).

Therefore, trypsin and collagenase were chosen as the final lysis drugs.

### 3.3. TEs can be taken up by thrombus-adjacent cells

To explore the functionality of the TEs, we labeled them using DiI and co-cultured them with thrombus-adjacent cells, that is, vascular smooth muscle cells, vascular endothelial cells, and cardiomyocytes. We suggested that these thrombus-adjacent cells could take up the TEs.

We used DiI staining, which is widely used to verify exosome uptake [24]. Subsequently, we used FITC labeling and staining for the cardiomyocyte marker cTnI and the smooth muscle cell marker αSMA, and the endothelial cell marker CD34. After labeling the cell nuclei with DAPI, the fluorescence of the cells was observed using a fluorescence microscope. Exosomes labeled with red fluorescence were completely internalized and taken up by cardiomyocytes (Fig 4A), smooth muscle cells (Fig 4B), and endothelial cells (Fig 4C). Our experiments suggested that these exosomes are likely to be functional, but their specific effects on cells adjacent to coronary thrombi require further investigation.

### 3.4. TEs promote HCM apoptosis, necrosis and autophagy

In this part of the experiment, we hypothesized that TEs regulate cardiomyocyte function. After co-culturing cardiomyocytes with TEs, we found that it can affected cardiomyocyte function by regulating cardiomyocyte apoptosis, autophagy, and ferroptosis.

To verify the function of coronary TEs in T1MI patients, different masses of TEs (50, 100, 200, 400, 800, 1600, 3200, 6400, and 12,800 ng) were co-cultured with HCM for 24 h. Subsequently, the CCK8 method was used to detect the cell activity in each group. As shown in Fig 5A, TE reached 12 800 ng(converted to a concentration of 128 ng/μL), the cell activity was significantly inhibited compared to the control group ($P<0.05$). Therefore, this concentration was chosen for our subsequent study.We also selected random healthy human serum-extracted exosomes (HSEs) as a negative control for TEs.

The groups were as follows: Control group (control), HSE group, AMI patient serum exosomes (ASE) group, and AMI patient TEs group. Exosomes from each group were co-cultured with HCM, the CCK8 results showed that the TE group had significantly inhibited cellular activity compared to that observed in the HSE group ($P<0.05$). Apoptosis was detected using flow cytometry (Fig 5C and 5D). We found that the TE group had a significantly increased cell necrosis rate compared to that in the control group ($P<0.05$). Western blot assay results are shown in Fig 5E–5G. The expression of GPX4, P62 were significantly reduced in the TE group compared to that in the control group ($P<0.05$), and the expression of caspase9, cleaved-

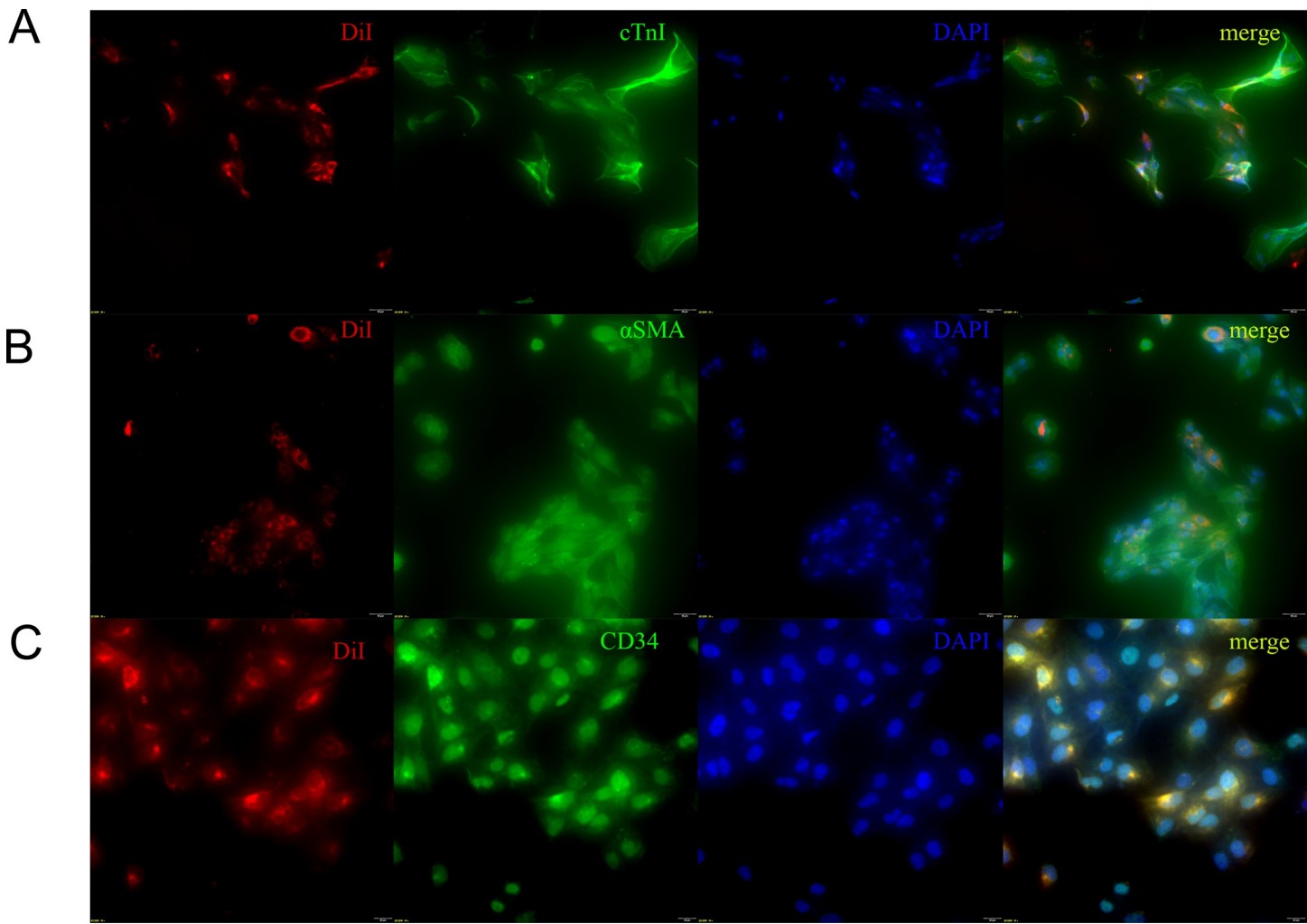

**Fig 4. The internalization and absorption of TEs by various thrombus-adjacent cells were identified by immunofluorescence (20X).** (A) Internalization and absorption of TEs by cardiomyocyte HCM. (B) Internalization and absorption of TEs by SMC. (C) Internalization and absorption of TEs by HUVEC.

caspase3 (P17) and LC3II/LC3I were significantly higher than that in the control group ($P<0.05$). In the intracellular ROS assay (Fig 5H and 5I), the mean fluorescence intensity of ROS was significantly higher in the TE group than that in the control group ($P<0.05$). Our findings suggest that the regulation of TE for cell necrosis might be related to ferroptosis, apoptosis and autophagy effects.

### 3.5. TEs inhibit HUVEC proliferation and migration and promote apoptosis

In this part of the experiment, we hypothesized that TEs regulate endothelial cell function in T1MI patients. CCK8 was uesd to verify the effect of TEs on HUVEC function. As shown in the results (Fig 6A), cell activity was significantly decreased in the ASE and TE groups compared to that in the control group ($P<0.05$). In contrast, this phenomenon was not observed in the HSE group ($P>0.05$). In the subsequent western blot assay (Fig 6B and 6C), the expression of cyclinD1 and PCNA was significantly decreased ($P<0.05$) and cleaved-caspase3 (P17) was significantly increased ($P<0.05$) in the TE group compared to that in the control group. This

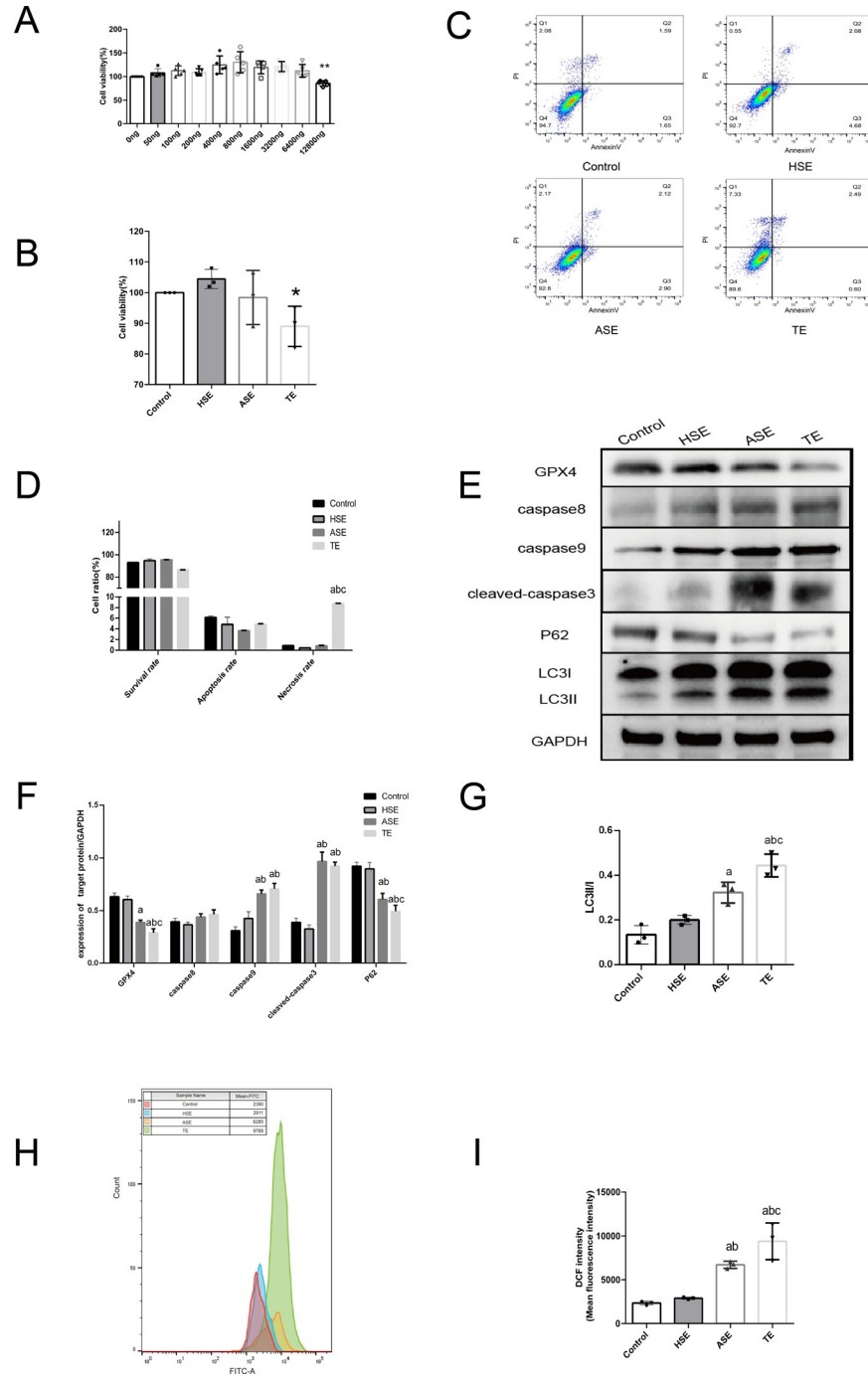

**Fig 5. Effect of TEs on HCM cell function.** (A) CCK8 assay of the effect of different concentrations of TEs on HCM cell activity. (B) CCK8 assay of the effect of three exosomes on HCM cell activity. (C) Flow cytometry assay of the effect of each group of exosomes on HCM cell apoptosis rate/necrosis rate (Annexin V/PI staining). (D) Flow cytometry assay Statistical plots. (E) Western blot detection of the effect of each group of exosomes on the expression of apoptosis, autophagy, and ferroptosis-related proteins in HCM cells. (F) Statistical plots of Western blot detection. (G) Statistical plots of grayscale values of LC3II/I detection. (H) Flow cytometry detection of each group of exosomes on ROS expression in HCM cells (DCFH-DA staining). (I) Statistical plots of mean fluorescence intensity of ROS. *$P<0.05$. **$P<0.01$. a:$P<0.05$ compared with Control group. b:$P<0.05$ compared with HSE group. c:$P<0.05$ compared with ASE group.

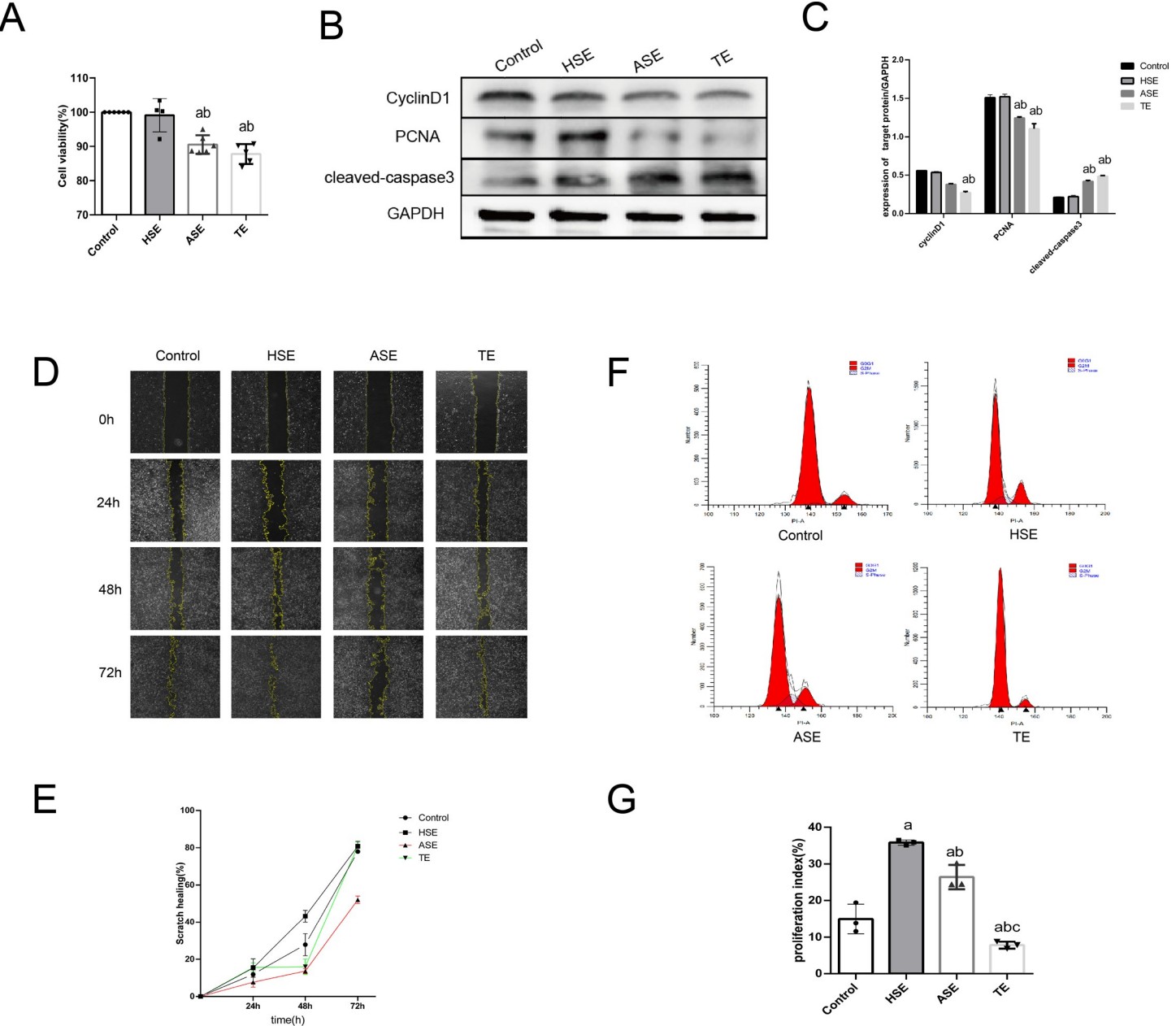

**Fig 6. Effect of TEs on HUVEC cell function.** (A) CCK8 assay of the effect of three exosomes on HUVEC cell activity. (B) western blot assay of various exosomes on HUVEC proliferation and apoptosis protein expression. (C) Statistical plot of western blot assay. (D) Migration capability was characterized by scratch wound healing assay after co-culture of each group of exosomes with HUVEC. (E) Statistical graph of cell scratch wound healing assay. (F) Flow cytometry detection of each cell cycle after co-culture of various exosomes with HUVEC. (G) Statistical graph of cell cycle PI values. a:$P<0.05$ compared with Control group. b:$P<0.05$ compared with HSE group. c:$P<0.05$ compared with ASE group. PI = (S+G2/M)/(G0/G1+S+G2/M)*100%.

indicated that the TEs inhibited cell proliferation and promoted apoptosis. To further investigate the effects of TEs on cell proliferation, wound healing, and cell cycle, assays were performed. As shown in Fig 6D and 6E, the cellular wound healing rate (%) was significantly decreased in the ASE group at 48h and 72h compared with that in the control group ($P<0.05$), whereas this was not observed in the TE and HSE groups ($P>0.05$). Subsequent cell cycle experiments showed that the PI of cells in the HSE and ASE groups increased significantly

($P$<0.05) compared with that in the control group, whereas the PI values in the TE group decreased significantly ($P$<0.05). Our study showed that TEs significantly inhibited HUVEC proliferation and migration, and promote apoptosis.

## 3.6. TEs promote SMC phenotypic transformation and proliferation

We hypothesized that TEs regulate SMC function in AMI patients.

CCK8 were examined the effect of exosomes on SMC activity. As shown in the results (Fig 7A), the activity of TE and HSE cells was significantly higher ($P$<0.05) than that of the control group, whereas that of the ASE group was significantly lower ($P$<0.05). In the subsequent western blot assay (Fig 7B and 7C), the expression of OPN, cyclinD1, and PCNA was significantly increased ($P$<0.05) and αSMA expression was significantly decreased ($P$<0.05) in the TE group compared to that in the control group. Subsequent cell cycle experiments showed (Fig 7D and 7E) that PI values were significantly higher in the TE group than in the control group ($P$<0.05). As shown in wound healing experiment results (Fig 7F and 7G), the cellular wound healing rate (%) was significantly decreased in the TE group at 12h and 24h compared to that in the control group ($P$<0.05), whereas this was not observed in the ASE and HSE groups ($P$>0.05). Our conclusions suggest that TEs promote the phenotypic conversion of SMC from contractile to secretory. It also promotes cell proliferation, but inhibits cell migration.

## 3.7. Identification of DELncRNAs

Given the pervasive involvement of long non-coding RNAs (lncRNAs) in a multitude of pathological processes and the dearth of knowledge regarding their role in cardiovascular disease, we subjected three exosomes (HSEs, ASEs and TEs) to lncRNA microarray sequence analysis. Three different samples were collected to verify the DELncRNAs among the different samples. based on the cut-off criteria of |logFC|<1.0, and $P$ <0.05. Compared with ASEs (Fig 8A and 8B), 1089 DELncRNAs were identified (517 upregulated and 572 downregulated). Compared with TEs (Fig 8C and 8D), 808 DEL ncRNAs were identified (780 upregulated and 28 downregulated). 780 upregulated and 28 downregulated genes). Compared with TEs, HSEs (Fig 8C and 8D) identified 302 DEL ncRNAs (265 upregulated and 37 downregulated). All data were subjected to |logFC| < 1.0 and $P$-value < 0.05, as cut-off criteria. We then considered the intersection of all upregulated and downregulated lncRNAs in each group separately. As shown in Fig 8G, the Venn diagram shows that among the upregulated lncRNAs, there were seven common lncRNAs, namely AC068418.2, AC010186.3, AL031430.1, BIG-lncRNA-1531, AC121333.1, CATG00000072758.1, and AL136526.1. In contrast, the downregulated lncRNAs were devoid of common lncRNAs (Fig 8F).

## 3.8. Association of DELncRNAs with autophagy and ferroptosis

We have previously found that TEs may be involved in ferroptosis and autophagy in thrombus-adjacent cells. To determine the relationship between the upregulated DELncRNAs expressed inside TEs, ferroptosis, and autophagy, we exported all mRNAs that could directly bind to these DELncRNAs from the RNAINTER database (see S1 Table). Unfortunately, two of the seven DELncRNAs (BIG-lncRNA-1531 and CATG00000072758.1) did not contain any information in the RNAINTER database. We did not find any bindable sites in the UCSC browser (http://genome.ucsc.edu/). Therefore, we evaluated only the remaining five DELncRNAs. As shown in the Venn diagram in Fig 9, AC068418.2, AC010186.3, AC121333.1, and AL136526.1 all involve both the autophagy and ferroptosis pathways (See Table 3 for specific results). All CGs contained TP53, TP63, and RELA (Table 3 for specific results). Only AL031430.1, did not have CGs in autophagy- and ferroptosis-related genes (Fig 9D).

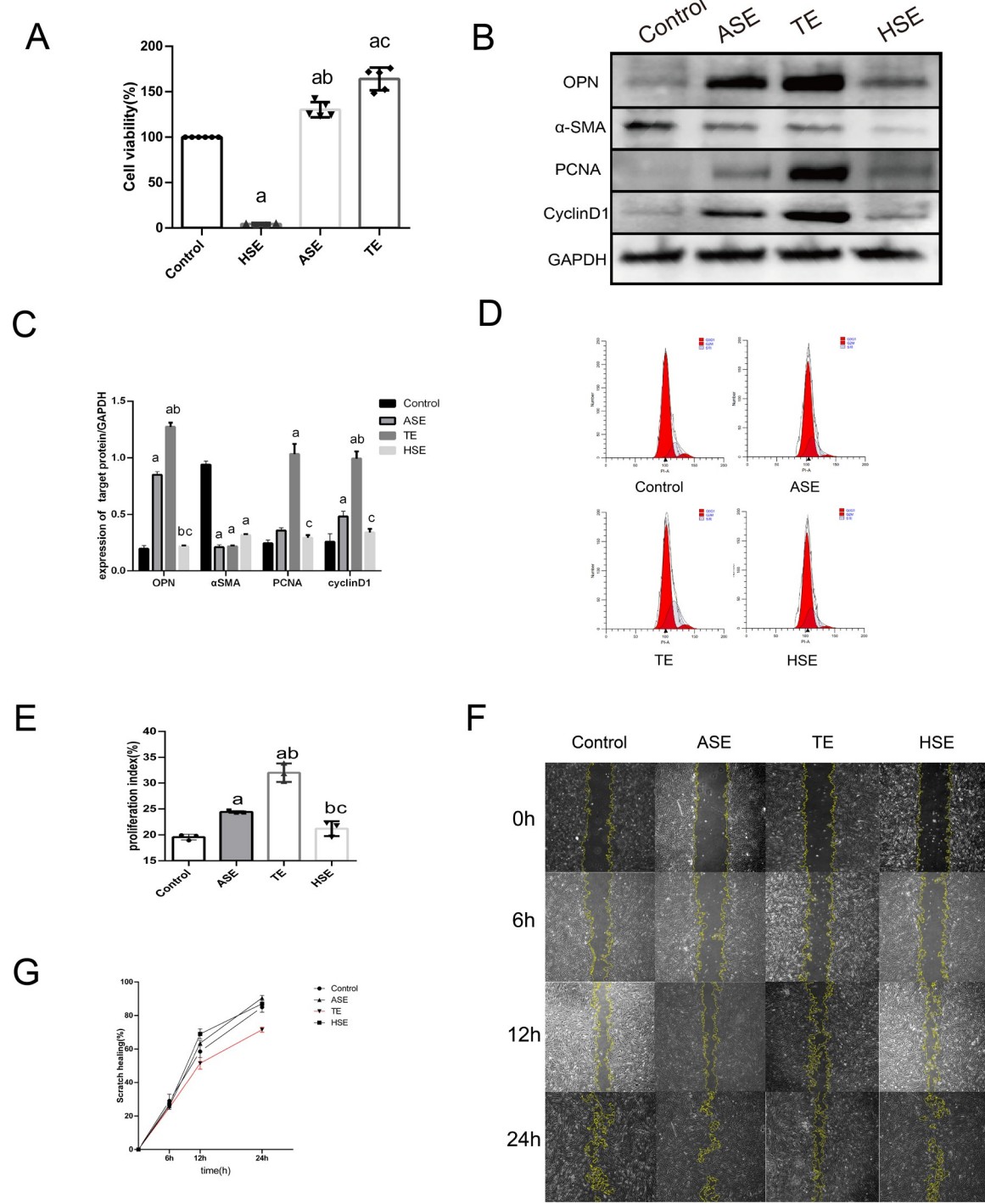

**Fig 7. Effect of thrombospondin exosomes on SMC cell function.** (A) CCK8 assay of the effect of three exosomes on SMC cell activity. (B) western blot assay of various exosomes on SMC cell proliferation as well as SMC phenotype transformation-related protein expression. (C) Statistical plot of western blot assay. (D) Flow cytometry detection of various exosomes and SMC co-culture after each cell cycle. (E) Statistical plot of cell cycle PI values. (F) Migration capability was characterized by scratch wound healing assay after co-culture of each group of exosomes with SMC. (G) Statistical graph of cell scratch wound healing assay. a:$P<0.05$ compared with Control group. b:$P<0.05$ compared with HSE group. c:$P<0.05$ compared with ASE group. PI = (S+G2/M)/(G0/G1+S+G2/M)*100%.

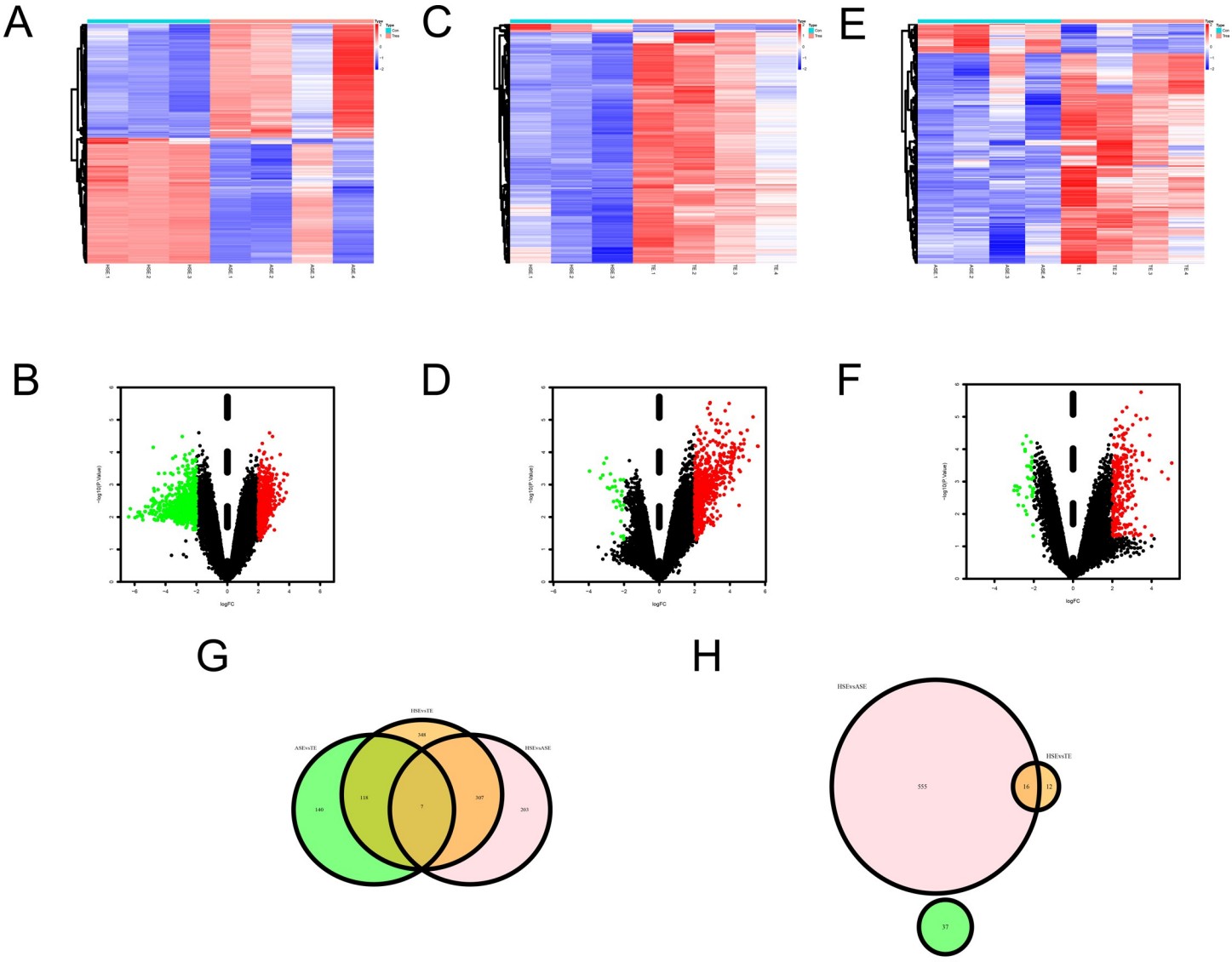

**Fig 8. Identification of DELncRNAs and CGs.** (A) Heat map showing the expression of DELncRNAs in healthy human serum extracted exosomes (HSEs) compared to AMI patient serum exosomes (ASEs). (B) Volcano plot of DELncRNAs between HSEs and ASEs. (C)Heat map showing the expression of DELncRNAs in HSEs compared to thrombus-derived exosomes (TEs). (D) Volcano plot of DELncRNAs between HSEs and TEs. (E)Heat map showing the expression of DELncRNAs in ASEs compared to TEs. (F) Volcano plot of DELncRNAs between ASEs and TEs. (G) Venn diagram showing intersected genes in three groups of mutually contrasting upregulated lncRNAs. (H)Venn diagram showing intersected genes in three groups of mutually contrasting downregulated lncRNAs.

## 4. Discussion

The fact that atherosclerotic plaques contain many vesicles was first reported back in 2007 by Leroyer et al. [5]. The role of these vesicles extends beyond their involvement in plaque formation; they also impact endothelial function and the accumulation of lipids and leukocytes in the endothelium. This, in turn, affects the stability of advanced plaques. Some studies have suggested that they may be involved in inducing thrombosis [6,7]. Evidence also suggests that these vesicles may be involved in the phenotypic transformation of smooth muscle cells through their enriched proteins and genetic material, as well as in the formation of foam cells after AS or AMI and the regulation of cardiomyocyte death [32]. In the present study, based

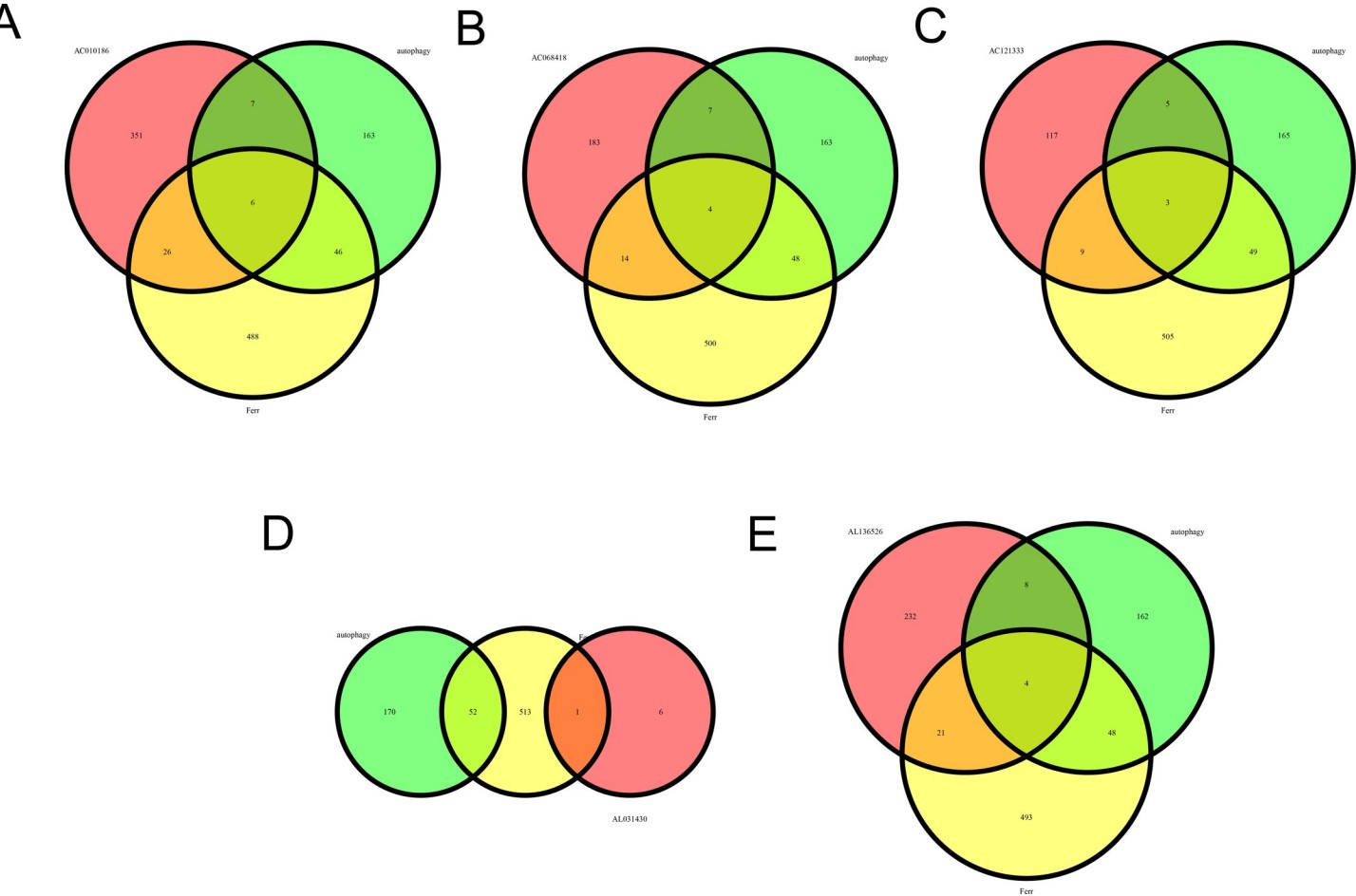

**Fig 9. Association of DELncRNAs with autophagy and ferroptosis.** (A) Venn diagram showing mRNAs that can bind directly to AC010186.3 and common genes (CGs) in autophagy and ferroptosis-related genes. (B) Venn diagram showing mRNAs that can bind directly to AC068418.2 and CGs in autophagy and ferroptosis-related genes. (C) Venn diagram showing mRNAs that can bind directly to AC121333.1 and CGs in autophagy and ferroptosis-related genes. (D) Venn diagram showing mRNAs that can bind directly to AL031430.1 and CGs in autophagy and ferroptosis-related genes. (E) Venn diagram showing mRNAs that can bind directly to AL136526.1 and CGs in autophagy and ferroptosis-related genes.

on the tissue exosome extraction method previously proposed by Vella's research group [16], namely the single collagenase lysis method, we compared various major thrombolytic drugs and different doses of trypsin and collagenase mixes. The final method of mixing type II collagenase with trypsin was screened and recommended based on the lysis efficiency of various methods and the proposed exosome acquisition rate. Currently, this technique has been

**Table 3. Intersecting genes common to each LncRNA with autophagy and ferroptosis.**

| LncRNA | CGs |
|---|---|
| AC068418.2 | RELA, TP63, HIF1A, TP53 |
| AC010186.3 | RELA, TP53, TP63, RB1, NFE2L2, HIF1A |
| AC121333.1 | RELA,TP63, TP53 |
| AL136526.1 | RELA,TP63,HIF1A,TP53 |

*CGs: Common Genes.*

awarded a national Patent No China (patent no. ZL202110554258.6). In the process of screening, we also found that high doses of trypsin may damage the membranes of exosomes, thus reducing their acquisition rate. We speculate that the underlying cause may be related to the disruption of the phospholipid bilayer structure by trypsin, which is consistent with the conclusion of previous studies showing that trypsin disrupts cell membranes [33].

In T1MI, cardiomyocyte necrosis is considered irreversible and rescuing deceased cardiomyocytes is currently the primary research focus. Several studies have shown that circulating exosomes are involved in local necrosis, autophagy, apoptosis, and ferroptosis in cardiomyocytes following a myocardial infarction [34]. In addition to the high risk of death in acute myocardial infarction, the subsequent heart failure (HF) can cause a drastic reduction in the patient's quality of life [35]. In previous studies, the phenomenon of no reflow due to vascular endothelial cell dysfunction after AMI and failure to establish collateral circulation were among the main causes of HF [36]. Evidence is available that exosomes or extracellular vesicles play pivotal roles in numerous physiological (immune response, cell-to-cell cooperation, and angiogenesis) and pathological (repair, inflammation, thrombosis/coagulation, atherosclerosis, and endothelial dysfunction) processes. HF development of HF is strongly associated with endothelial dysfunction, microvascular inflammation, alterations in tissue repair, and cardiac and vascular remodeling [37]. Our preliminary study examined the fundamental functional effects of these TEs on vascular endothelial cells, vascular smooth muscle cells, and cardiomyocytes. The findings of our study indicate that TEs may play a role in the stimulation of cardiomyocyte apoptosis and ferroptosis. This finding suggests that, in addition to the vascular blockage caused by the thrombus itself after AMI, which causes downstream blood supply disorders, the enriched exosomes within the thrombus also may exacerbate apoptosis and necrosis of cardiomyocytes.

Phenotypic conversion of smooth muscle cells has an important influence on the regulation of vascular function after AMI and myocardial fibrosis. Regarding studies on smooth muscle phenotypic switching, numerous pieces of research demonstrate that inhibiting smooth muscle phenotypic switching after acute myocardial infarction effectively reduces the formation of scar tissue and promotes improvement in cardiac function In addition to damage to cardiomyocytes, phenotypic conversion of smooth muscle cells has an important influence on the regulation of vascular function after AMI and myocardial fibrosis. Regarding studies on smooth muscle phenotypic switching, numerous pieces of research demonstrate that inhibiting smooth muscle phenotypic switching after acute myocardial infarction effectively reduces the formation of scar tissue and promotes improvement in cardiac function [38]. The findings of our study indicate that TEs may contribute to the promotion of necrosis, apoptosis, and ferroptosis in cardiomyocytes. Additionally, they hinder the proliferation and migration of vascular endothelial and smooth muscle cells. It is possible that these factors may contribute to the increased myocardial cell necrosis and vascular injury that occurs following the onset of an acute myocardial infarction.

Autophagy and ferroptosis are of significant importance in the context of cardiomyocytes following a myocardial infarction [39,40]. A substantial body of evidence has corroborated the assertion that long non-coding RNAs (lncRNAs) exert a pivotal influence in this process [40]. In the present study, we used lncRNA microarrays to explore the specific mechanisms by which TEs regulate autophagy and ferroptosis in cardiomyocytes. Our findings suggested that only five lncRNAs were co-expressed in the three exosomes (HSEs, ASEs, and TEs) with other possible gene- or protein-binding sites, namely AC068418.2, AC010186.3, AL031430.1, AC121333.1, and AL136526.1. Using bioinformatic techniques, we analyzed the genes to which these lncRNAs may bind and found that almost all of them could be involved in the regulation of ferroptosis or autophagy. Among them, four, AC068418.2, AC010186.3, AC121333.1, and

AL136526.1, are involved in the regulation of both ferroptosis and autophagy. Notably, they may directly bind to TP53, TP63, and RELA, thus regulating autophagy and ferroptosis in cells. Interestingly, among the four lncRNAs studied, only AC010186.3 [41] was hypothesized to be involved in regulating autophagy in ovarian cancer cells. The study of the remaining lncRNAs is still in a gap. Moreover, TP53, TP63 and RELA are classical autophagy-related genes or proteins [42,43], which also have extremely important roles in ferroptosis [44–46]. Therefore, our findings suggest a new direction for research on TEs, autophagy, ferroptosis, and lncRNAs.

With the rise in noncellular therapeutic techniques for exosomes, the relationship between exosomes and myocardial infarction (MI) has received increasing attention. Exosomes secreted by various cells have been shown to regulate biological functions of various cells in the heart after MI [47]. Our study also suggested that exosomes contained in the thrombus can be internalized and absorbed by vascular cells and cardiomyocytes, implying that these exosomes have specific functions. There is no doubt that the internalized uptake of exosomes by various cells is a prerequisite for their biological functions [48], and the widespread use of exosomes as novel drug carriers at home and abroad is based on this property [49]. The transport properties, biocompatibility, and low immunogenicity of exosomes make them excellent candidates for clinical tools and drug delivery vesicles [50]. For example, Adam et al. [51] used a cardiac-homing peptide to target exosomes derived from cardiac stem cells, resulting in increased homing of these exosomes to cardiomyocytes after acute MI, ultimately improving myocardial function. Exosomes enriched in p53-responsive miR-34a, miR-192, and miR-194 have been demonstrated to aid the prediction of HF after MI [52]. By combining previous research with the results of the present study, we aimed to offer new directions for clinical research. Specifically, we propose two goals: first, the potential use of TEs as a more precise predictor of cardiac impairment following AMI in future studies, and second, a more comprehensive understanding of the sources of these TEs so that targeted preventive therapy can be administered prior to T1MI onset, targeting our predicted biomarkers, including TP53, TP63, and RELA.

This study had some limitations. For instance, our analysis of the functions of TEs is quite superficial, and we have not thoroughly examined the pertinent results garnered from the application of bioinformatics techniques. Furthermore, in this study, we primarily employed established cell lines for cellular investigations, rather than primary cells. This may also introduce some bias into the results of this study. We plan to explore these aspects in future studies. Overall, the main objective of this study is to propose a new concept for TEs. At the same time, we hope to point out a new research direction for our discipline, namely, the effect of TEs on the function of cardiac spectrum cells after AMI. We hope that this will lay a new theoretical foundation for future basic and noncellular therapies for AMI.

## 5. Conclusions

This study proposed the concept of intrathrombotic exosomes. We extracted TEs for the first time and verified their functions. TEs may regulate ferroptosis and autophagy in thrombus-adjacent cells through the enrichment of certain lncRNAs. In conclusion, our results suggest TEs as a new target and research direction for the treatment of HF after T1MI.

## Supporting information

**S1 Table. Autophagy and ferroptosis related genes.**
(XLSX)

**S1 Raw images.**
(PDF)

## Acknowledgments

We sincerely thank the Department of Cardiology of Guizhou Provincial People's Hospital (Guizhou Cardiovascular Hospital) for providing the thrombus. We would like to thank Editage (www.editage.cn) for the English language editing.

## Author Contributions

**Conceptualization:** Youfu He, Bo Wang.

**Data curation:** Youfu He, Bo Wang, Yu Qian, Debin Liu.

**Formal analysis:** Youfu He, Bo Wang, Yu Qian, Debin Liu.

**Funding acquisition:** Youfu He, Qiang Wu.

**Investigation:** Youfu He, Bo Wang, Qiang Wu.

**Methodology:** Youfu He, Bo Wang, Qiang Wu.

**Resources:** Youfu He.

**Software:** Yu Qian, Debin Liu.

**Supervision:** Yu Qian, Debin Liu.

**Validation:** Qiang Wu.

**Writing – original draft:** Youfu He, Bo Wang, Qiang Wu.

**Writing – review & editing:** Youfu He, Bo Wang, Qiang Wu.

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
