## [Decision Letter · Decision Letter 0]

3 Sep 2024

PONE-D-24-31496Extraction of coronary thrombus-derived exosomes from patients with acute myocardial infarction and its effect on the function of adventitial cellsPLOS ONE

Dear Dr. He,

Thank you for submitting your manuscript to PLOS ONE. After careful consideration, we feel that it has merit but does not fully meet PLOS ONE’s publication criteria as it currently stands. Therefore, we invite you to submit a revised version of the manuscript that addresses the points raised during the review process.

We look forward to receiving your revised manuscript.

Kind regards,

Vinay Kumar, Ph.D.

Academic Editor

PLOS ONE

Journal Requirements:

1. When submitting your revision, we need you to address these additional requirements. Please ensure that your manuscript meets PLOS ONE's style requirements, including those for file naming. The PLOS ONE style templates can be found at https://journals.plos.org/plosone/s/file?id=wjVg/PLOSOne_formatting_sample_main_body.pdf and https://journals.plos.org/plosone/s/file?id=ba62/PLOSOne_formatting_sample_title_authors_affiliations.pdf 2. Thank you for stating the following financial disclosure: "This work was supported by the [Guizhou Provincial Health and Wellness Commission] under Grant [gzwkj 2021-102]; The National Natural Science Foundation of China (No. 82260084); and Guizhou Provincial Science and Technology Agency Project (Qian Ke He Foundation ZK [2023] General 217)." Please state what role the funders took in the study.  If the funders had no role, please state: "The funders had no role in study design, data collection and analysis, decision to publish, or preparation of the manuscript." If this statement is not correct you must amend it as needed. Please include this amended Role of Funder statement in your cover letter; we will change the online submission form on your behalf. 3. PLOS ONE now requires that authors provide the original uncropped and unadjusted images underlying all blot or gel results reported in a submission’s figures or Supporting Information files. This policy and the journal’s other requirements for blot/gel reporting and figure preparation are described in detail at https://journals.plos.org/plosone/s/figures#loc-blot-and-gel-reporting-requirements and https://journals.plos.org/plosone/s/figures#loc-preparing-figures-from-image-files. When you submit your revised manuscript, please ensure that your figures adhere fully to these guidelines and provide the original underlying images for all blot or gel data reported in your submission. See the following link for instructions on providing the original image data: https://journals.plos.org/plosone/s/figures#loc-original-images-for-blots-and-gels.   In your cover letter, please note whether your blot/gel image data are in Supporting Information or posted at a public data repository, provide the repository URL if relevant, and provide specific details as to which raw blot/gel images, if any, are not available. Email us at plosone@plos.org if you have any questions.

Reviewers' comments:

Reviewer's Responses to Questions

**Comments to the Author**

1. Is the manuscript technically sound, and do the data support the conclusions?

Reviewer #1: Partly

Reviewer #2: No

2. Has the statistical analysis been performed appropriately and rigorously? 

Reviewer #1: N/A

Reviewer #2: Yes

3. Have the authors made all data underlying the findings in their manuscript fully available?

Reviewer #1: Yes

Reviewer #2: Yes

4. Is the manuscript presented in an intelligible fashion and written in standard English?

Reviewer #1: No

Reviewer #2: No

5. Review Comments to the Author

Reviewer #1: The manuscript "Extraction of coronary thrombus-derived exosomes from patients with acute myocardial infarction and its effect on the function of adventitial cells" describes a method to isolate exosomes from coronary thrombus. This study is important because Type I acute myocardial infarction (T1MI) has a very high morbidity and mortality rate across the world and the role of coronary thrombus derived exosomes is not well studied.

In this manuscript the authors tested different concentrations and incubation time for collagenase and trypsin for the extraction of exosomes from coronary thrombus. They were able to isolate vesicles which could be exosomes after centrifugation. They then performed several cell biological experiments to test the effect of TEs on thrombus-adjacent cells. Finally they performed a microarray analysis for lncRNA's from thrombus isolated exosomes and compared them with HSEs and ASEs. They found several differentially expressed lncRNA's among the three exosomes. This is a well planned and executed study but I have a few concerns

Major Concerns

1. Throughout the manuscript the authors have over analysed the findings. This becomes more problematic in the discussion section where the authors talk in great deal about the deferentially expressed lncRNA which they identified only by one microarray experiment and were not validated in an independent experiment like Real Time PCR.

2. The use of strong language and over interpretation of results can also be seen in Lines 505-506 and 514-515.

3. In line 514 the authors say "our findings confirmed". They should rather use a language such as our findings suggest. No one experiment can confirm anything in biology and a series of experiments preferentially done by different group of people are required before stating that something is confirmed.

4. Several parts of the manuscript needs to be rewritten especially the discussion.

Minor Concerns

1. There are several grammatical and writing errors for example line 359, 488, 369 etc

2. A brief explanation of vesicles is needed on line 61. Authors jump from exosomes to vesicles making the reader wonder whether they are talking about the same thing or if they are different.

Reviewer #2: In the present study, the authors investigate the role of thrombus-derived exosomes (TEs) in Type I acute myocardial infarction (T1MI), focusing on their effects on adjacent cells and their potential as therapeutic targets. The authors optimize the protocol for extracting TEs and examined their impact on cardiomyocytes and endothelial cells. The findings show that TEs promote necrosis, autophagy, and ferroptosis in human cardiomyocytes, while inhibiting the proliferation and migration of endothelial cells. Additionally, TEs stimulate smooth muscle cell proliferation and migration. Bioinformatics analysis highlighted five lncRNAs (AC068418.2, AC010186.3, AL031430.1, AC121333.1, and AL136526.1) that regulate autophagy and ferroptosis.

However, the following issues must be addressed.

1. As noted in the introduction, exosomes contain a complex mixture of cellular components. The authors could have provided a more comprehensive analysis of the exosomal composition, such as through proteomics or genomics, rather than focusing solely on lncRNAs. The rationale for selecting only lncRNAs should be clearly explained in the manuscript.

2. Additionally, the authors could have performed GO term or KEGG pathway analyses to explore the broader functional implications of the identified lncRNAs.

3. The study is limited to cell lines, and the inclusion of experiments using isolated primary cardiomyocytes or endothelial cells would have strengthened the findings. This limitation should be acknowledged in the discussion.

4. It is unclear whether the exosomes used for cell line treatments were pooled from five AMI patients. The methods section lacks clarity and should be updated with detailed information on the exosome isolation and pooling procedures.

5. Figures 2F and 3F might be better presented in the supplementary material, perhaps in a table format.

6. Lines 369-373 contain repeated sentences and should be revised for clarity. The authors should also carefully review the manuscript for English language accuracy.

7. Finally, is there a possibility that the authors isolated exosomes from patients who underwent AMI treatment? If so, it would be insightful to assess the activity of these TEs on HCM cell lines.

6. PLOS authors have the option to publish the peer review history of their article (what does this mean?). If published, this will include your full peer review and any attached files.

Reviewer #1: No

Reviewer #2: **Yes: **Suman Asalla

---

## [Author Response · Author response to Decision Letter 0]

14 Oct 2024

List of Responses

Dear Editors and Reviewers:

Thank you for your letter and for the reviewers’ comments concerning our manuscript entitled “Extraction of coronary thrombus-derived exosomes from patients with acute myocardial infarction and its effect on the function of adventitial cells”. (ID: PONE-D-24-31496).

Those comments are all valuable and very helpful for revising and improving our paper, as well as the important guiding significance to our researches. We have studied reviewer’s comments carefully and have made revision which marked in red in the paper. We have tried our best to revise our manuscript according to the comments. Attached please find the revised version, which we would like to submit for your kind consideration. The main corrections in the paper and the responds to the reviewer’s comments are as flowing:

Reviewer #1: The manuscript "Extraction of coronary thrombus-derived exosomes from patients with acute myocardial infarction and its effect on the function of adventitial cells" describes a method to isolate exosomes from coronary thrombus. This study is important because Type I acute myocardial infarction (T1MI) has a very high morbidity and mortality rate across the world and the role of coronary thrombus derived exosomes is not well studied.In this manuscript the authors tested different concentrations and incubation time for collagenase and trypsin for the extraction of exosomes from coronary thrombus. They were able to isolate vesicles which could be exosomes after centrifugation. They then performed several cell biological experiments to test the effect of TEs on thrombus-adjacent cells. Finally they performed a microarray analysis for lncRNA's from thrombus isolated exosomes and compared them with HSEs and ASEs. They found several differentially expressed lncRNA's among the three exosomes. This is a well planned and executed study but I have a few concerns.

Major Concerns

1. Throughout the manuscript the authors have over analysed the findings. This becomes more problematic in the discussion section where the authors talk in great deal about the deferentially expressed lncRNA which they identified only by one microarray experiment and were not validated in an independent experiment like Real Time PCR.

Response: Thank you very much to reviewer 1 for recognising our study. Regarding the validation of lncRNA, we have actually done a lot of qPCR validation. For example, in Figure A of this reply letter, it is the qPCR validation of the differentially expressed gene top20. Of course, the purpose of this study was mainly to confirm the existence of thrombus-derived exosomes (TE) and to explore their role on thrombus clinical cells. Therefore, after completing the RNA microarray analysis, we did not delve further into its specific mechanism.

2. The use of strong language and over interpretation of results can also be seen in Lines 505-506 and 514-515.

Response: Many thanks to reviewer 1 for your patience and correction. We have re-examined this manuscript and corrected many places that were inappropriately described.

3. In line 514 the authors say "our findings confirmed". They should rather use a language such as our findings suggest. No one experiment can confirm anything in biology and a series of experiments preferentially done by different group of people are required before stating that something is confirmed.

Response: Many thanks to reviewer 1 for your patience and correction. Regarding the word confirmed, we also found inappropriate use of the word elsewhere in the manuscript. It has now been corrected in the manuscript.

4. Several parts of the manuscript needs to be rewritten especially the discussion.

Response: Thank you so much for your feedback. After receiving the review comments, our team members did find a lot of inappropriate things after checking and reviewing again. It has now been reworked for revision.

Minor Concerns

1. There are several grammatical and writing errors for example line 359, 488, 369 etc

Response: Indeed, as reviewer 1 said, in many places our wording is not very accurate and there is a suspicion of exaggerating the results. We fully accept the criticisms of the reviewers and editors and thank you for your comments. After careful consideration, many inappropriate places have been reworked.

2. A brief explanation of vesicles is needed on line 61. Authors jump from exosomes to vesicles making the reader wonder whether they are talking about the same thing or if they are different.

Response: Many thanks to reviewer 1 for his patience and correction. Indeed we have overlooked these basic details. Considering that exosome itself also belongs to a kind of extracellular vesicle, so we added a part of explanation and definition for exosome.

Reviewer #2: In the present study, the authors investigate the role of thrombus-derived exosomes (TEs) in Type I acute myocardial infarction (T1MI), focusing on their effects on adjacent cells and their potential as therapeutic targets. The authors optimize the protocol for extracting TEs and examined their impact on cardiomyocytes and endothelial cells. The findings show that TEs promote necrosis, autophagy, and ferroptosis in human cardiomyocytes, while inhibiting the proliferation and migration of endothelial cells. Additionally, TEs stimulate smooth muscle cell proliferation and migration. Bioinformatics analysis highlighted five lncRNAs (AC068418.2, AC010186.3, AL031430.1, AC121333.1, and AL136526.1) that regulate autophagy and ferroptosis.

However, the following issues must be addressed.

1. As noted in the introduction, exosomes contain a complex mixture of cellular components. The authors could have provided a more comprehensive analysis of the exosomal composition, such as through proteomics or genomics, rather than focusing solely on lncRNAs. The rationale for selecting only lncRNAs should be clearly explained in the manuscript.

Response: Thanks a lot for this professional review. Because of the complexity of the mechanism of lncRNA, that is why we chose to do the lncRNA microarray analysis separately. Also, the microarray analysis we did was not exclusively for lncRNAs, but also for many mRNAs, and we wrote a separate article to discuss this part. Of course, as reviewer 2 said, we did lack the explanation of this part. Therefore, we have made some additions in RESULT 3.7 of the manuscript.

2. Additionally, the authors could have performed GO term or KEGG pathway analyses to explore the broader functional implications of the identified lncRNAs.

Response: Reviewer 2 is a very knowledgeable professor of academics, so please allow me to express my high regard to you. As we said earlier, our microarray sequencing analyses also included a lot of mRNA results. So combining these mRNA and lncRNA results, we did perform a lot of KEGG and Pathway analyses. A portion of the results can be seen in Figure B. Considering that the position of this article is to confirm that the substance TE does exist and can have a certain regulatory effect on thrombus adventitial cells. So our exploration in this part would like to be a bit more superficial.

3. The study is limited to cell lines, and the inclusion of experiments using isolated primary cardiomyocytes or endothelial cells would have strengthened the findings. This limitation should be acknowledged in the discussion.

Response: Thanks again to reviewer 2 for your comments. Your rigour is something we should respect and learn from. We did use some common cell lines for our study. So this is an important shortcoming of our study. Combined with your comments, we have added in the limitation section comments.

4. It is unclear whether the exosomes used for cell line treatments were pooled from five AMI patients. The methods section lacks clarity and should be updated with detailed information on the exosome isolation and pooling procedures.

Response: Please forgive us for our oversight. In this study, practically all exosomes used for cellular studies were obtained from these three T1MI patients. Taking your comments into account, we have refined the description in the Methods section.

5. Figures 2F and 3F might be better presented in the supplementary material, perhaps in a table format.

Response: Please allow me to explain the rationale for this data. For exosome identification, we routinely need to use these three methods, i.e. western blot, electron microscopy and Nanoparticle tracking analysis (NTA). Figure 2F and Figure 3F are both essential NTA data in the exosome identification repertoire. In this field, we need to see the maximum peaks in addition to the distribution of particle sizes to understand the purity of the extracted exosomes. So we ultimately chose to keep this part of the data in order to demonstrate the credibility of our data as much as possible.

6. Lines 369-373 contain repeated sentences and should be revised for clarity. The authors should also carefully review the manuscript for English language accuracy.

Response: We are very grateful to reviewer 2 for your careful correction. Please excuse our carelessness, the repetitive description of grouping in this section has been corrected.

7. Finally, is there a possibility that the authors isolated exosomes from patients who underwent AMI treatment? If so, it would be insightful to assess the activity of these TEs on HCM cell lines.

Response: Indeed, the TEs as well as the ASEs we extracted were from patients undergoing emergency PCI in our hospital. As described in our methods, all thrombi and sera were extracted during emergency PCI. Considering that TE is really something that is overlooked by all of us researchers in the cardiovascular field, we take it very seriously. We also think it's a very valuable subject to study. We are currently doing everything we can to explore the association of TE with AMI as a disease. We hope that our research can provide a new line of treatment for AMI, especially T1MI as a disease. Therefore, we also very much hope that the reviewers and editors will give us this opportunity to present our study to the cardiovascular field researchers worldwide through this journal. We also hope that more researchers will join the TE study to provide more theoretical basis for the treatment of T1MI patients in the future.

We tried our best to improve the manuscript and made some changes in the manuscript. These changes will not influence the content and framework of the paper. And here we did not list the changes but marked in red in revised paper. In addition, we will be happy to edit the paper further, based on helpful comments from the reviewers. 

We appreciate for Editors/Reviewers’ warm work earnestly, and hope that the correction will meet with approval.

Once again, thank you very much for your comments and suggestions.

Figure A. RT-PCR analysis of DElncRNA expression in each group of exosomes

a, compared with the HE group, p < 0.05; b, compared with the SE group, p < 0.05

HE, serum exosomes from healthy individuals; SE, serum exosomes from patients with AMI; TE, coronary thrombus-derived exosomes from patients with AMI

Figure B. Bioinformatics analysis of lncRNAs enriched in coronary thrombus-derived exosomes (a) Heat map showing the expression of DElncRNAs in TEs compared with those in healthy human serum exosomes. Each row in the heat map represents a single gene and each column represents a tissue sample. The color scale, ranging from blue (low expression) to red (high expression), represents the raw Z-score mRNA intensity values; (b) Volcano plot of DElncRNAs between TEs and healthy human serum exosomes. The red nodes represent up-regulated DEGs and the green nodes represent down-regulated DEGs; (c) Heat map showing the expression of DEmRNAs in TEs compared with those in healthy human serum exosomes; (d) Volcano plot of DEmRNAs between TEs and healthy human serum exosomes; (e,f) Circle chart and bubble diagram of gene ontology (GO) analysis of DEmRNAs showing enriched biological processes, cell components, and molecular functions; (g,h) Circle chart and bar chart of Kyoto Encyclopedia of Genes and Genomes (KEGG) pathway analysis of targeted genes showing enriched signaling pathways.

HE, serum exosomes from healthy individuals; SE, serum exosomes from patients with AMI; TE, coronary thrombus-derived exosomes from patients with AMI

---

## [Decision Letter · Decision Letter 1]

28 Oct 2024

Extraction of coronary thrombus-derived exosomes from patients with acute myocardial infarction and its effect on the function of adventitial cells

PONE-D-24-31496R1

Dear Dr. He,

We’re pleased to inform you that your manuscript has been judged scientifically suitable for publication and will be formally accepted for publication once it meets all outstanding technical requirements.

Kind regards,

Vinay Kumar, Ph.D.

Academic Editor

PLOS ONE

Additional Editor Comments (optional):

Reviewers' comments:

Reviewer's Responses to Questions

**Comments to the Author**

1. If the authors have adequately addressed your comments raised in a previous round of review and you feel that this manuscript is now acceptable for publication, you may indicate that here to bypass the “Comments to the Author” section, enter your conflict of interest statement in the “Confidential to Editor” section, and submit your "Accept" recommendation.

Reviewer #1: All comments have been addressed

Reviewer #2: All comments have been addressed

2. Is the manuscript technically sound, and do the data support the conclusions?

Reviewer #1: Yes

Reviewer #2: Yes

3. Has the statistical analysis been performed appropriately and rigorously? 

Reviewer #1: Yes

Reviewer #2: Yes

4. Have the authors made all data underlying the findings in their manuscript fully available?

Reviewer #1: Yes

Reviewer #2: Yes

5. Is the manuscript presented in an intelligible fashion and written in standard English?

Reviewer #1: Yes

Reviewer #2: Yes

6. Review Comments to the Author

Reviewer #1: The authors have addressed all the comments in a satisfactory manner. They have toned down the language in the discussion section according to the suggestion.

Reviewer #2: (No Response)

7. PLOS authors have the option to publish the peer review history of their article (what does this mean?). If published, this will include your full peer review and any attached files.

Reviewer #1: No

Reviewer #2: No

---

## [Editor Report · Acceptance letter]

7 Nov 2024

PONE-D-24-31496R1 

PLOS ONE

Dear Dr. He, 

I'm pleased to inform you that your manuscript has been deemed suitable for publication in PLOS ONE. Congratulations! Your manuscript is now being handed over to our production team.

Kind regards, 

on behalf of

Dr. Vinay Kumar 

Academic Editor

PLOS ONE